# Antispasmodic Effect of *Valeriana pilosa* Root Essential Oil and Potential Mechanisms of Action: Ex Vivo and In Silico Studies

**DOI:** 10.3390/pharmaceutics15082072

**Published:** 2023-08-02

**Authors:** Roberto O. Ybañez-Julca, Ricardo Pino-Ríos, Iván M. Quispe-Díaz, Daniel Asunción-Alvarez, Edwin E. Acuña-Tarrillo, Elena Mantilla-Rodríguez, Patricia Minchan-Herrera, Marcelo A. Catalán, Liz Zevallos-Escobar, Edison Vásquez-Corales, Osvaldo Yáñez, Wilfredo O. Gutiérrez-Alvarado, Julio Benites

**Affiliations:** 1Facultad de Farmacia y Bioquímica, Universidad Nacional de Trujillo, Trujillo 13011, Peru; iquispe@unitru.edu.pe (I.M.Q.-D.); hasuncion@unitru.edu.pe (D.A.-A.); eacunat@unitru.edu.pe (E.E.A.-T.); amantilla@unitru.edu.pe (E.M.-R.); pminchan@unitru.edu.pe (P.M.-H.); 2Química y Farmacia, Facultad de Ciencias de la Salud, Universidad Arturo Prat, Casilla 121, Iquique 1100000, Chile; rpinorios@unap.cl; 3Instituto de Estudios de la Salud, Universidad Arturo Prat, Casilla 121, Iquique 1100000, Chile; 4Instituto de Fisiología, Facultad de Medicina, Universidad Austral de Chile, Valdivia 5090000, Chile; marcelo.catalan@uach.cl; 5Escuela de Farmacia y Bioquímica, Universidad Católica Los Ángeles de Chimbote, Chimbote 02801, Peru; lzevallose@uladech.edu.pe (L.Z.-E.); evasquezc@uladech.edu.pe (E.V.-C.); 6Facultad de Ingeniería y Negocios, Universidad de las Américas, Santiago 7500000, Chile; oyanez@udla.cl; 7Facultad de Farmacia y Bioquímica, Universidad Nacional de la Amazonía Peruana, Iquitos 16001, Peru; wilfredo.gutierrez@unapiquitos.edu.pe

**Keywords:** *Valeriana pilosa*, antispasmodic effect, essential oil, molecular docking

## Abstract

Infusions of *Valeriana pilosa* are commonly used in Peruvian folk medicine for treating gastrointestinal disorders. This study aimed to investigate the spasmolytic and antispasmodic effects of *Valeriana pilosa* essential oil (VPEO) on rat ileum. The basal tone of ileal sections decreased in response to accumulative concentrations of VPEO. Moreover, ileal sections precontracted with acetylcholine (ACh), potassium chloride (KCl), or barium chloride (BaCl_2_) were relaxed in response to VPEO by a mechanism that depended on atropine, hyoscine butylbromide, solifenacin, and verapamil, but not glibenclamide. The results showed that VPEO produced a relaxant effect by inhibiting muscarinic receptors and blocking calcium channels, with no apparent effect on the opening of potassium channels. In addition, molecular docking was employed to evaluate VPEO constituents that could inhibit intestinal contractile activity. The study showed that α-cubebene, β-patchoulene, β-bourbonene, β-caryophyllene, α-guaiene, γ-muurolene, valencene, eremophyllene, and δ-cadinene displayed the highest docking scores on muscarinic acetylcholine receptors and voltage-gated calcium channels, which may antagonize M_2_ and/or M_3_ muscarinic acetylcholine receptors and block voltage-gated calcium channels. In summary, VPEO has both spasmolytic and antispasmodic effects. It may block muscarinic receptors and calcium channels, thus providing a scientific basis for its traditional use for gastrointestinal disorders.

## 1. Introduction

Gastrointestinal disorders are related to motility disturbances, visceral hypersensitivity, altered mucosal and immune functions, gut microbiota dysbiosis, and impaired regulation by the central nervous system [1]. Gastrointestinal dysfunction is associated with significant global healthcare costs [2,3] and decreased quality of life [4]. Existing synthetic antispasmodic drugs may cause unpleasant side effects such as dizziness, blurred vision, fatigue, and dry mouth. Therefore, discovering new molecules with antispasmodic properties is an important goal for the pharmaceutical industry [5].

Medicinal plants are complex mixtures of compounds that have served as sources of drugs for centuries. Approximately half of the current pharmaceutical therapies are derived from medicinal plants [6].

Little is known about the effects of essential oils on the intestinal microbiota [7], which has been demonstrated to modulate intestinal motility [8,9,10,11]. Interestingly, short-chain fatty acids, mainly produced by intestinal bacteria, modulate intestinal motility [10].

*Valeriana pilosa* R & P. is a plant belonging to the genus Valeriana and distributed in the Andean region. In Peru, it is known as “Valeriana”, “Coche coche”, ‘‘Valeriana de paramo’’, ‘‘Ornamo’’, or “Babilla”. Vertical rhizome and attached roots from *Valeriana pilosa* are widely used as antispasmodic, relaxing, anti-inflammatory, and sleep-promoting agent [12,13].

Our continuous interest in the volatile components of plant species from Chilean and Peruvian Andean highland communities has been the focus of our chemical and biological research projects [14,15,16,17,18,19].

In this study, we report spasmolytic and antispasmodic activities of VPEO on intestinal contractile activity. The VPEO-mediated effect appears to be mediated by muscarinic receptor antagonism, calcium channel blockade, and potassium channel opening, which may explain the traditional use of the plant in gastrointestinal disorders. Additionally, we performed in silico molecular docking analyses of the major constituents in VPEO and found that some molecules have the potential to bind to M_2_ muscarinic acetylcholine receptors (M_2_R), M_3_ muscarinic acetylcholine receptors (M_3_R), and voltage-gated calcium channels (Ca_v_1.2).

## 2. Materials and Methods

### 2.1. Chemicals, Drugs, and Solutions

Acetylcholine hydrochloride (ACh), barium chloride (BaCl_2_), ethylenediaminetetraacetic acid (EDTA), calcium chloride (CaCl_2_), glibenclamide, D-glucose, magnesium chloride (MgCl_2_), potassium chloride (KCl), potassium dihydrogen phosphate (KH_2_PO_4_), sodium bicarbonate (NaHCO_3_), sodium chloride (NaCl), sodium phosphate monobasic (NaH_2_PO_4_), and verapamil were purchased from Sigma-Aldrich (St. Louis, MO, USA) and Merck (Peruana S. A, Ate, Lima, Perú).

### 2.2. Plant Material

Roots (250 g) of *Valeriana pilosa* R & P. were collected in January 2022 in the Community of San Juan de Corralpampa at 3500 m above sea level, in the province of Hualgayoc, Department of Cajamarca, Perú. They were then subjected to hydrodistillation for 3 h, using a Clevenger-type apparatus. The oil obtained was dried over anhydrous sodium sulfate. Afterward, it was filtered and stored under protection at +4 °C for further analysis and testing [20].

### 2.3. Gas Chromatography Analysis (GC) and Gas Chromatography–Mass Spectrometry (GC–MS) of Valeriana Pilosa Essential Oil (VPEO)

All chemicals were of analytical reagent grade; they were obtained from Sigma-Aldrich-Fluka (St. Louis, MO, USA) and Merck (Darmstadt, Germany), and were used as supplied. Chromatographic analysis of VPEO was performed using a Perkin Elmer Clarus 600 gas chromatograph, following to the procedure reported by Benites et al. [19].

GC–MS analysis of VPEO was carried out as previously reported [20]. Briefly, analyses were performed on a Perkin Elmer Clarus 600 gas chromatograph, consisting of a DB-1 fused-silica column (30 m × 0.25 mm i.d., film thickness 0.25 uM; J & W Scientific, Folsom, CA, USA), and interfaced with a Perkin Elmer Clarus 600T mass spectrometer (software version 4.1, Perkin Elmer, Shelton, CT, USA). The injector and oven temperatures were as follows: transfer line temperature, 280 °C; ion source temperature, 220 °C; carrier gas, helium, adapted to a linear rate of 30 cm/s; split ratio, 1:40; ionization energy, 70 eV; scan range, 40–300 m/z; and scan time, 1 s. The components were identified by comparing their retention indices relative to C_9_–C_21_ n-alkane indices and GC–MS spectra from the mass spectra library and commercial sample standards.

### 2.4. Animals

The experiments in this study were performed following the procedures of the American Veterinary Medical Association (AVMA) [21] and the Ethics Committee of Pharmacy and Biochemistry Faculty of the National University of Trujillo (COD.N°: P 012-19/CEIFYB). Twenty male rats (8–10 weeks old *Rattus norvegicus* Holtzman, 170–200 g) were housed in cages (22–25 °C, 12 h light/dark cycles) with *ad libitum* access to standard rat chow (Molinorte S.A.C., Trujillo, Peru) and water.

### 2.5. Preparation of Rat Ileum

Animals were sacrificed by cervical dislocation. A portion of the ileum (2.5 cm), without considering the 10 cm nearest to the ileocecal valve, was removed and placed into a petri dish containing Tyrode’s solution (concentrations in mM): NaCl 136.9; KCl 2.68; CaCl_2_ 1.8; MgCl_2_ 1.05; NaHCO_3_ 11.9; NaH_2_PO_4_ 0.42; and D-glucose 5.55 [22]. Ileum fragments were placed into an isolated organ chamber filled with 25 mL of Tyrode´s solution. The chamber was kept at 37 °C continuously gassed with a mixture of 95% O_2_ and 5% CO_2_ (pH 7.4). Resting tension was fixed at 1 g. The experimental data were recorded using a Power Lab 26T system (AD-Instruments Pty Ltd., Bella Vista, NSW, Australia) with the LabChart 8 program for Windows (Colorado Springs, CO, USA).

### 2.6. Ex Vivo Experimental Protocol

#### 2.6.1. Effect of VPEO on the Basal Tone of Rat Ileum

The contractility of the rat ileum in response to VPEO was assessed through cumulative dose–response experiments. A series of seven distinct concentrations of VPEO (1, 10, 100, 250, 500, 750, and 1000 μg/mL) were administered in 5 min intervals [22].

#### 2.6.2. Spasmolytic Effect in Precontracted Rat Ileum

Neurotropic spasm was induced using acetylcholine (ACh), whereas musculotropic spasm was induced using highly concentrated potassium chloride (KCl)- or barium chloride (BaCl_2_)-containing solutions [22]. Isolated ileal sections were treated with ACh (10^−5^ M), KCl (60 mM), or BaCl_2_ (5 mM) for 10 min or until a stable contractile plateau (plateau) was reached. Then, different VPEO concentrations (1, 10, 100, 250, 500, 750, and 1000 μg/mL) were sequentially added to the tissue chamber.

#### 2.6.3. Antispasmodic VPEO Effect in the Dose–Response Curves of ACh, KCl, and BaCl_2_

The effect of VPEO on the contractibility in response to ACh was assessed as follows: dose–response experiments (ACh, 10^−10^ to 10^−3^ M) before and after VPEO (250 and 500 μg/mL) administration were performed in the same experiment.

Similar experiments to those described above but contracting the ileal sections with KCl (10^−4^ to 10^−1^ M) or BaCl_2_ (10^−8^ to 10^−2^ M) were performed.

Dose–response data were fitted to the standard Hill’s function using Graphpad.

#### 2.6.4. Role of Extracellular Ca^2+^ Influx in the Intestinal VPEO-Mediated Relaxation

In addition to the Tyrode´s solution, a calcium-free solution was prepared for this experiment. The solution had the following composition (concentrations in mM): KCl 50; NaCl 91.04; MgCl_2_ 1.05; NaHCO_3_ 11.87; NaH_2_PO_4_ 0.41; glucose 5.55; and EDTA 0.1 [23]. Initially, the tissue was stabilized in normal Tyrode´s solution and then replaced with a Ca^2+^-free Tyrode´s solution. Then, 10 min after the addition of calcium-free solution, the intestinal sections were contracted with ACh 10^−5^ M, followed by the addition of increasing CaCl_2_ concentrations (0.1 mM; 0.3 mM; 0.6 mM; and 1 mM). Then, the tissue was washed with normal Tyrode’s solution for at least 10 min, followed by the addition of VPEO (250 and 500 μg/mL) for 20 min [22].

After VPEO incubation in normal Tyrode’s solution, intestinal sections were incubated with VPEO-supplemented calcium-free solution, then contracted with ACh 10^−5^ M, followed by increasing CaCl_2_ concentrations (0.1 mM; 0.3 mM; 0.6 mM, and 1 mM).

The tension was re-adjusted in the middle of the experiment to 1 g when necessary [22].

#### 2.6.5. Effect of VPEO on Muscarinic Receptors

To analyze the role of VPEO in muscarinic receptor activity, experiments were carried out in the absence/presence of atropine (a non-selective muscarinic antagonist) [24], hyoscine butylbromide (M_2_–M_3_ blockers) [25], or solifenacin (selective M_3_ blocker) [26]. First, the tissue was stabilized for 1 h and then washed in 15 min intervals (4–5 times) with Tyrode’s solution. The tension was adjusted to 1 g when necessary. Then, ileal sections were treated with increasing doses of VPEO (1, 10, 100, 250, 500, 750, and 1000 μg/mL). Afterward, the sections were washed 4–5 times in 15 min intervals, and the tension was adjusted to 1 g. The tissue was pre-incubated for 20 min with 1.0 μM atropine, 1.0 μM hyoscine butylbromide, or 1.0 μM solifenacin, and then different VPEO concentrations were added (1–1000 μg/mL).

#### 2.6.6. Effects of VPEO on Voltage-Gated Calcium Channel

To determine whether the effect of VPEO was related to the inhibition of voltage-gated calcium channel, experiments were performed in the absence and presence of verapamil (a voltage-gated calcium channel blocker). Initially, the tissue was stabilized for 1 h and then washed in 15 min intervals (4–5 times). The tension was adjusted again to 1 g if necessary. The tissue was pre-incubated with verapamil (1.0 μM) [27] for 20 min. Then, different VPEO concentrations (1, 10, 100, 250, 500, 750, and 1000 μg/mL) were added in intervals of 5 min.

#### 2.6.7. Effects of VPEO on Potassium Channel Blockers

The role of K⁺ channels in VPEO-induced relaxation was investigated by pre-incubating the ileum rat for 20 min with two K^+^ channel blockers, namely, glibenclamide [28] (ATP sensitive K⁺ channel blocker), and barium chloride [29] (an inward rectifier K⁺ channel blocker). The tissue was stabilized for 1 h and then washed in 15 min intervals (4–5 times). The tension was adjusted again to 1 g if necessary. The tissue was pre-incubated with glibenclamide (10 μM) and barium chloride (1 mM) for 20 min. Then, VPEO was added (1, 10, 100, 250, 500, 750, and 1000 μg/mL) in intervals of 5 min for each successive dose, and the response was recorded.

### 2.7. In Silico Studies

Molecular modelling analysis of compounds **1–47** (see the SMILES in Appendix A) of VPEO [20] to M_2_ Muscarinic Acetylcholine Receptor [30], M_3_ Muscarinic Acetylcholine Receptor [31], and Ca_v_1.2 L-type voltage-gated calcium channel [32] were performed using AutoDock (v 4.2.1), AutoDock Vina (v 1.0.2) [33], and AutoDockTools packages [34]. The crystal structures considered for these docking studies had the following PDB Codes: 3UON (M_2_ Muscarinic Acetylcholine Receptor), 4DAJ (M_3_ Muscarinic Acetylcholine Receptor), and 5V2P (L-type voltage-gated calcium channel). Data was obtained from the Protein Data Bank [35]. The three-dimensional coordinates of all structures were optimized using MOPAC2016 software by the PM6-D3H4 semi-empirical method [36,37]. The crystal structures were treated with Schrödinger’s Protein Preparation Wizard [38]; polar hydrogen atoms were added, nonpolar hydrogen atoms were merged, and charges were assigned. Docking was treated as rigid and performed using the empirical free energy function and Lamarckian Genetic Algorithm provided by AutoDock Vina [39]. The grid map dimensions were 20 × 20 × 20 Å^3^. The center of the binding site were the following coordinates for each of the proteins studied (Table 1).

All other parameters were set to their default values as defined by AutoDock Vina. Dockings were repeated 10 times, with the space search exhaustiveness set to 100. The best interaction binding energy (kcal·mol^−1^) was selected for evaluation. Docking results in 3D representations were obtained using Discovery Studio [40] 3.1 (Accelrys, San Diego, CA, USA) molecular graphics system. The co-crystallized ligands were removed from their proteins and saved separately in the PDB format, which was used for redocking their respective protein active domains to validate our docking methodology [20].

Ligand Efficiency (*LE*), Binding Efficiency Index (*BEI*), Lipophilic Ligand Efficiency (*LLE*), and Non-Covalent Interaction Index (NCI) were calculated according to a previously reported procedure [20].

### 2.8. Statistical Analysis

GraphPad Prism 8.0.2 software (San Diego, CA, USA) was used for the statistical analyses. Non-linear regression analysis (three-parameter Hill function) was generated to compare dose–response curves. Two-way ANOVA analysis followed by the Bonferroni post hoc test was used to evaluate the significance between different groups.

## 3. Results

### 3.1. Spasmolytic Activity of VPEO on Rat Ileum

VPEO significantly decreased the muscle tone of the rat ileum at 100 µg/mL (70.63 ± 4.82% reduction compared to the control condition; *p* < 0.01) (Figure 1A). VPEO-mediated effect was dose-dependent, where 750 and 1000 µg/mL doses showed a higher effect compared to the 100 µg/mL dose (*p* < 0.05).

The spasmolytic activity was determined by measuring the relaxation induced by VPEO in ileal sections precontracted with (a) 10^−5^ M ACh (muscarinic agonist) (Figure 1B), (b) 60 mM KCl (depolarizes the smooth muscle cell membrane and opens voltage-dependent Ca^2+^ channels) (Figure 1C), and (c) 5 mM BaCl_2_ (a non-selective blocker of the current rectifying potassium channels; K_IR_) (Figure 1D).

VPEO significantly relaxed, in a dose-dependent manner, precontracted intestinal segments with ACh, KCl, and BaCl_2_. For instance, 100 µg/mL VPEO relaxed ileal sections precontracted with ACh, KCl, and BaCl_2_ by 54.62 ± 5.7%, 60.88 ± 6.2%, and 73.89 ± 2.2%, respectively.

### 3.2. Antispasmodic Activity of VPEO on Rat Ileum

#### 3.2.1. Role of Muscarinic Acetylcholine Receptors

Experiments were performed to determine whether VPEO modifies ACh-evoked intestinal contractions in the intact rat ileal tissue. Initially, ileum contractions were recorded before and after VPEO treatment (Figure 2A). VPEO at a dose of 500 μg/mL significantly reduced (*p* < 0.05) the ACh-induced contraction (10^−7^ M ACh): 40.5 ± 8.45% (control) vs. 18.4 ± 1.96% (VPEO); Figure 2B,B.1. Similarly, in the presence of 1 μM atropine (a non-selective muscarinic receptor antagonist), the contractile effect of ACh (10^−7^ M) was suppressed (4.6 ± 1.19%, *p* < 0.001), as shown in Figure 2B.1. Moreover, increasing concentrations of ACh in the presence of VPEO significantly reduced ACh-induced contractions in the rat ileum (Figure 2B.1–B.4). Interestingly, in the Figure 2B, the nonlinear regression analysis of the dose–response curve shows that the pEC_50_ values for ACh in the presence of 250 µg/mL (6.22 ± 0.09) and 500 µg/mL (5.75 ± 0.08) of VPEO were significantly different to that obtained under control conditions (6.53 ± 0.11).

#### 3.2.2. Effect of VPEO on Intestinal Sections Precontracted with KCl or BaCl_2_

Figure 3A shows the effect of different KCl concentrations on ileum contractions. To study the antispasmodic activity, intestinal strips from wild-type rats were pre-incubated with the essential oil. VPEO significantly reduced the contraction induced by 10^−2^ M KCl: 71.46 ± 8.93% (*p* < 0.05) and 38.58 ± 9.74% (*p* < 0.001) compared to the control in the presence of 250 µg/mL and 500 µg/mL VPEO, respectively (Figure 3B.1). Similar results were observed for 10^−1.5^ M KCl: 62.82 ± 6.03% (*p* < 0.01) and 59.03 ± 10.46% (*p* < 0.01) compared to the control in the presence of 250 µg/mL and 500 µg/mL VPEO, respectively (Figure 3B.2). In Figure 3B, the nonlinear regression analysis of the dose–response curve shows that the sensitivity to KCl (pEC_50_) in the presence of 250 µg/mL (2.12 ± 0.25) and 500 µg/mL (1.74 ± 0.29) VPEO was not significantly different from that obtained under control condition (2.19 ± 0.22).

On the other hand, high concentrations of BaCl_2_ (10^−8^–10^−2^ M) block not only inward rectifier potassium channels (K_IR_), but also voltage-gated potassium channels (VGKCs). In Figure 3C, the dose–response curves of ileum contractions induced by BaCl_2_ in the presence and absence of VPEO are recorded. Pre-incubation with VPEO significantly reduced the contraction induced by 10^−4^ M BaCl_2_: 23.75 ± 2.47% (250 µg/mL VPEO; *p* < 0.01) and 31.18 ± 11% (500 µg/mL VPEO; *p* < 0.05; Figure 3D.1); and 10^−3^ M BaCl_2_: 79.5 ± 1% (250 µg/mL VPEO; *p* < 0.01) and 73 ± 5.2% (500 µg/mL VPEO; *p* < 0.05, Figure 3D.2) compared to the value obtained under control condition. In Figure 3D, the nonlinear regression analysis of the dose–response curve shows that the sensitivity to BaCl_2_ (pEC_50_) in the presence of 250 µg/mL (3.35 ± 0.06) and 500 µg/mL (3.68 ± 0.18) VPEO was not significantly different from that obtained under control condition (3.95 ± 0.36).

### 3.3. Determination of the Mechanisms of Action Underlying the VPEO Effect

#### 3.3.1. Effect of VPEO on Muscarinic Receptors

After observing that VPEO reduced the effect of ileum contractions induced by ACh, we studied how VPEO affected the blockade of muscarinic receptors. The intestinal strips were pre-incubated with atropine (a non-selective muscarinic antagonist), hyoscine butylbromide (M_2_–M_3_ muscarinic antagonist), and solifenacin (M_3_ selective muscarinic antagonist), at doses of 1 μM; then, concentrations of VPEO were added.

Figure 4A,B shows tissues pre-incubated with atropine and hyoscine butylbromide, respectively, at a dose of 1 μM. In ileal tissues pre-incubated with 1 μM atropine (Figure 4A) or 1 μM hyoscine butilbromide (Figure 4B), the relaxant effect of 100 μg/mL VPEO was significantly (*p* < 0.05) reduced (0.67 ± 5.54% and 0.63 ± 10.55%) compared to ileal tissues that were not pre-incubated with antagonists (29.37 ± 4.82%). At the highest dose of VPEO (1000 μg/mL), the relaxation was more significant against atropine 1 μM (18.33 ± 7.67%, *p* < 0.01) than in hyoscine butylbromide 1 μM (30 ± 5.4%, *p* < 0.05) compared to the control (57.15 ± 5.84%). On the other hand, the relaxant dose–response of VPEO in the presence of solifenacin 1 μM decreased compared with that of the control, albeit not significantly (*p* > 0.05), as shown in Figure 4C.

These results indicate that the blockade of M_3_ and M_2_ receptors mediate the relaxant effect of VPEO.

#### 3.3.2. Extracellular Ca^2+^ Dependence of VPEO Effect

The relation between the relaxant effect of VPEO (250 and 500 μg/mL) and extracellular calcium was assessed. The contraction induced by the cumulative concentration of extracellular Ca^2+^ ions (0 to 1 mM) in ileum rats precontracted with ACh (10^−5^ M) maintained in Ca^2+^-free Tyrode’s solution (containing 0.1 mM EDTA) in the presence and absence of VPEO was determined (Figure 5A). The cumulative addition of calcium ions in Ca^2+^-free Tyrode increased in the contraction of ileal sections in a concentration-dependent manner. Pre-treatment with VPEO at concentrations of 500 µg/mL (171 ± 29.1% vs. 249.1 ± 23.8 control; *p* < 0.01) significantly decreased the contraction induced by CaCl_2_ 0.6 mM. However, the concentration–response curve produced by the different concentrations of calcium ions was significantly decreased in the presence of VPEO compared to the control (Figure 5B).

The relaxant effect produced by VPEO was decreased in the presence of verapamil 1 μM (*p <* 0.05, Figure 5C). For instance, relaxation induced by VPEO in ileal sections treated with 100 μg/mL VPEO was 29.37 ± 4.82 vs. 6.73 ± 1.16, control vs. verapamil, respectively. Taken together, the results suggest that essential oils might decrease contraction by reducing the entry of calcium ions into smooth muscle cells produced by extracellular calcium.

#### 3.3.3. Effect of Two K⁺ Channel Blockers on the Relaxation in Precontracted Ileum

The antispasmodic effect of VPEO in the rat ileum pre-incubated for 20 min with two K⁺ channel blockers, glibenclamide (an ATP-sensitive K⁺ channel blocker, 3 μM) and barium chloride (inward rectifier K⁺ channel blocker, 1 mM) was studied. The relaxation induced by VPEO in the rat ileum was unaffected by glibenclamide (Figure 6A) or BaCl_2_ (Figure 6B) treatment. Therefore, we conclude that *Valeriana pilosa* root essential oil does not induce opening of potassium channels in rat ileum.

### 3.4. Molecular Docking and Ligand Efficiency Analysis of VPEO

We have recently described the chemical composition and antioxidant activity of the essential oil from *Valeriana pilosa* roots (VPEO). A total of 47 compounds were identified, where sesquiterpene hydrocarbons, monoterpene hydrocarbons, oxygenated monoterpenes, and oxygenated sesquiterpenes were the major constituents. Within the sesquiterpenes, the major constituents of VPEO were α-patchoulene (5.8%), α-humulene (6.1%), seychellene (7.6%), and patchoulol (20.8%), as well as spathulenol, T-cadinol, and γ-cadinol as minor constituents [20].

Molecular docking was performed to identify possible VPEO constituents that could inhibit some proteins involved in intestinal contractile activity. Based on the results obtained in the current study, we selected M_2_ and M_3_ Muscarinic Acetylcholine Receptor and Ca_v_1.2 (L-type voltage-gated calcium channel). Table 2 shows a heat map of intermolecular docking energy values of 47 VPEO compounds. The values appear in a three-color scheme (red–yellow–green), where red represents the most stable binding energies and green represents the least stable ones. Table 2 shows a clear trend of compounds acting as putative inhibitors of the M_2_ and M_3_ muscarinic acetylcholine receptors.

Based on the information shown in Table 2, M_2_ and M_3_ muscarinic acetylcholine receptors appeared to be the best protein targets for some VPEO constituents, as shown by their intermolecular docking energy (∆*E_binding_*), score normalization of the binding energy based on the number of non-hydrogen atoms (*IE_norm__.binding_*), and Ligand Efficiency (*LE*) values. Indeed, using the weighted arithmetic mean values (related to % of composition, see Appendix A) shown in Table 3 obtained for all compounds, binding values were used for the analysis of M_2_ muscarinic acetylcholine receptor: ∆*E_binding_*, *LE*, and *IE_norm_*_.*binding*_ values were −7.67, 0.55, −2.06 kcal·mol^−1^, and the M_3_ muscarinic acetylcholine receptor, the ∆*E_binding_*, *LE*, and *IE_norm.binding_* were −8.03, 0.58, and −2.15 kcal·mol^−1^, respectively, while, in comparison with the Ca_v_1.2 (L-type voltage-gated calcium channel) protein target, the ∆*E_binding_*, *LE*, and *IE_norm.binding_*, binding values were −5.11, 0.37, and −1.37 kcal·mol^−1^. Therefore, these results show that the M_3_ muscarinic acetylcholine receptor, as a target protein, is probably involved in the effects of VPEO, which means that they might be involved in the anti-spasmodic activities of VPEO compounds. 

Molecular docking analyses, K_d_ values, Ligand Efficiency (*LE*), Binding Efficiency Index (*BEI*), Lipophilic Ligand Efficiency (*LLE*), and normalization of the binding energy score based on the number of non-hydrogen atoms (*IE_norm.binding_*) are summarized in Table 4 for the top 12 (M_2_ Muscarinic Acetylcholine Receptor) and 11 (M_3_ Muscarinic Acetylcholine Receptor) compounds, of which the most abundant compounds found in the VPEO are presented. The results showed that approximately 60% of the VPEO compounds acted as potential blockers of M_2_ and M_3_ Muscarinic Acetylcholine Receptors. In contrast, the others showed higher values and were considered weak blockers with low activity for the other protein target Ca_v_1.2 (L-type voltage-gated calcium channel).

VPEO compounds obtained from molecular docking (see Table 4) displaying more binding potential to M_2_ Muscarinic Acetylcholine Receptor were β-patchoulene, β-caryophyllene, γ-muurolene, and eremophyllene, while for the M_3_ Muscarinic Receptor were α-cubebene, β-patchoulene, β-bourbonene, β-caryophyllene, α-guaiene, α-humulene, γ-muurolene, valencene, and δ-cadinene, all obtained ∆*E_binding_* values (<−9.0) and *IE_norm, binding_* values (<−2.2), representing those with the highest interaction with residues close to the active site. In addition, the binding energies were evaluated for the major VPEO compounds, including natural sesquiterpenes such as α-patchoulene (5.8%), α-humulene (6.1%), seychellene (7.6%), and patchoulol (20.8%).(Detailed values of the interactions between individual compounds of the essential oil and the mentioned targets can be found Appendix A)

Table 5 lists the non-covalent interactions present in the M_2_-ligand complex. The four best compounds, β-patchoulene, β-caryophyllene, γ-muurolene, and eremophyllene, identified by molecular docking presented weak Van der Waals-type and π–alkyl interactions (see Table 5 and Figure 7) with the binding site of M_2_, where the most representative residues were Ala191, Ala194, Cys429, Phe181, Trp155, Trp400, Tyr104, and Tyr403. On the other hand, for the major VPEO compounds (see Table 5 and Figure 7), α-patchoulene, α-humulene, seychellene, and patchoulol had similar interactions of weak Van der Waals-type and π–alkyl interactions with the binding site of M_2_, where the most representative residues of these interactions are Ala191, Ala194, Phe181, Trp155, Trp400, Tyr104, and Tyr403. However, patchoulol has a hydrogen bridge interaction between the hydroxyl group and Asn404, forming a favorable interaction in the binding of this compound by M_2_.

To extend the analysis, we focused on all possible protein–ligand interactions in the set of the best eight (identified by molecular docking) and the major VPEO compounds that interact with the M_3_ Muscarinic Acetylcholine Receptor binding site (Table 6). These interactions define an interaction framework for the ligands in the analyzed set.

Figure 8 shows the non-covalent interactions present in the M_3_-ligand complex. The five best compounds, such as α-cubebene, β-patchoulene, β-bourbonene, β-caryophyllene, α-guaiene, γ-muurolene, valencene, and δ-cadinene, found by molecular docking present weak Van der Waals-type and π–alkyl interactions with the aromatic binding site of M_3_, where the most representative residues are Tyr506, Tyr148, Phe239, Trp503, Tyr506, Trp199, and Tyr529. On the other hand, the major constituents of the VPEO as α-patchoulene, α-humulene, seychellene, and patchoulol, had similar interactions of weak Van der Waals-type and π–alkyl interactions with the aromatic binding site of M_3_, where the most representative residues of these interactions are Ala235, Cys532, Val155, Trp503, Tyr506, and Tyr529. However, patchoulol has a hydrogen bridge interaction between the hydroxyl group and Asn507, favoring the potential binding of this compound to M_3_.

## 4. Discussion

Essential oils are gaining interest due to their intricate chemical composition and diverse pharmacological effects. Among these actions, the antispasmodic effect is well known, although further research is required to better understand the cellular and molecular mechanisms of action [41]. Therefore, we studied the antispasmodic effect of *Valeriana pilosa* root essential oil (VPEO) and its underlying mechanisms of action using the isolated rat ileum ex vivo model. Our findings reveal that VPEO displays spasmolytic and antispasmodic effects in a dose-dependent manner at concentrations ranging from 100 to 1000 μg/mL. The relaxant activity of VPEO in the isolated rat ileum was reversible after washing, indicating that the inhibition observed was not attributable to intestinal damage caused by the oil’s interaction with cell lipid bilayers [42].

Smooth muscle excitation and contraction in response to acetylcholine (ACh) release from autonomic nerves primarily involve the activation of muscarinic acetylcholine receptors (mAChRs) in the gastrointestinal tract and many other visceral organs [43]. The mAChR family comprises five molecularly distinct subtypes: M_1_–M_5_ [44]. Smooth muscle mAChRs consist of M_2_ and M_3_ subtypes, with M_2_ being predominant (M_2_:M_3_ = 3–5:1) [43,45], although all five mAChR subtypes have been detected in the gastrointestinal smooth muscle at the mRNA level [46].

Activation of mAChRs triggers multiple biochemical and electrical signaling events that result in muscle contraction [47]. Our results demonstrated that VPEO, at a concentration of 250 μg/mL, reduced tonic contraction in ACh-precontracted tissues. Additionally, preincubation of intestinal tissues with VPEO (250 μg/mL) significantly reduced the pEC_50_ of the tissue to ACh, as evidenced by a rightward shift in the dose–response curve. These initial findings suggest that VPEO blocks muscarinic receptors involved in rat ileum contraction. To explore this hypothesis, we evaluated whether the presence of muscarinic antagonists such as atropine, hyoscine butylbromide, and solifenacin would attenuate the relaxing effect of VPEO. Our results showed that solifenacin, a selective blocker of M_3_ muscarinic receptors, slightly reduced the relaxation induced by VPEO, but not significantly. In contrast, hyoscine butylbromide, an M_2_–M_3_ muscarinic blocker, significantly reduced the relaxing effect of VPEO. Moreover, when the non-selective muscarinic blocker atropine was employed, the relaxing effect of VPEO on muscle tone in the rat ileum was further reduced. Traditional studies using various mAChR antagonists have suggested that the M_3_ subtype primarily mediates contraction in visceral smooth muscles, while the contribution of the M_2_ subtype remains less clear [48,49]. However, other investigations have demonstrated that the M_2_ subtype modulates contraction, at least in part, by inhibiting cyclic AMP (cAMP)-dependent relaxation [43] and regulating smooth muscle ion channel activity [50,51,52]. Nevertheless, the precise mAChR subtype that mediates the contractile response remains largely unknown. Our findings indicate that, in addition to blocking M_3_ receptors, the antispasmodic effect of VPEO also involves modulation of M_2_ receptors. Consequently, both experimental and theoretical outcomes reveal the impact of VPEO on M_2_ and M_3_ muscarinic receptors.

There is limited research available on the effects of essential oils on cholinergic receptors expressed in smooth muscle, and a few studies have been published [6]. Molecular interactions between essential oil terpene compounds, such as vanillin, pulegone, eugenol, carvone, carvacrol, carveol, α-terpineol, thymol, thymoquinone, menthol, menthone, menthone, limonene, and nicotinic cholinergic receptors have been reported [53,54,55].

High concentrations of potassium ions (hypermolar KCl) cause tonic contraction of smooth intestinal muscles by a mechanism that depends on extracellular calcium influx [56,57]. Our study demonstrated that increasing concentrations of VPEO reduced contractions induced by 60 mM KCl. Additionally, when the tissue was pre-incubated for 20 min with the essential oil, the dose–response curves in response to KCl were significantly reduced. The action of VPEO can be attributed to its major component, patchouli alcohol (20.8%), a tricyclic sesquiterpene that has displayed similar results in other types of smooth muscle, such as the aorta [58] and rat corpus cavernosum [59]. However, the synergistic effects of other VPEO components, such as α-patchoulene (5.8%), α-humulene (6.1%), and seychellene (7.6%), at lower concentrations cannot be rejected.

Furthermore, it was confirmed that the relaxation induced by VPEO was reduced in the presence of verapamil, a voltage-gated calcium channel blocker. These results are consistent with similar studies conducted on the rhizome of *Valeriana hardwickii* Wall, which caused a rightward shift in calcium concentration-response curves in the rabbit jejunum, similar to that caused by verapamil [60,61]. The findings from our study on VPEO suggest the presence of an antispasmodic effect potentially mediated through calcium channel blockade.

It has been shown that ATP-sensitive potassium (K_ATP_) channels regulate intestinal contractility [28]. These channels are heterooctameric proteins composed of pore-forming subunits of the inwardly rectifying K^+^ channel (K_IR_) and subunits of the sulfonylurea regulatory receptor (SUR). The data demonstrate decreased intrinsic basal contractility in the small intestine and colon due to increased basal K_ATP_ channel activity, which can be inhibited by glibenclamide, an ATP-sensitive K⁺ channel blocker [62], and BaCl_2_, an inward rectifier K⁺ channel blocker [63,64]. We observed that VPEO reduced BaCl_2_-induced contractions, shifting the dose–response curve to the right, suggesting that VPEO might activate K^+^ channels, similar to the crude extract of *Valeriana wallichii*, which displays antispasmodic activity mediated by activation of the K_ATP_ channel [65]. To confirm these results, tissues were pre-incubated with glibenclamide and BaCl_2_, and the relaxing effect of VPEO on basal tone was evaluated. Our results showed that the activation of K_ATP_ and K_IR_ channels did not produce the relaxation exhibited by VPEO in the rat ileum.

The monoterpenoid group, including compounds such as α-pinene, myrcene, limonene, *p*-cymene, 1,8-cineole, camphor, carvone, α-humulene, and spathulenol isolated from terrestrial plants, represents the chemical group with the highest number of antispasmodic compounds [66], which are also present in VPEO.

It is important to mention that several studies have shown that phytochemical analysis of essential oils from numerous plants can induce relaxation in smooth muscle across various animal species, which is often attributed to their antioxidant activity and ability to enhance gut and barrier function in animals [6,49,67]. Terpenes have been demonstrated to enhance gut microbiota, such as D-limonene [68,69], suggesting that this could be a possible explanation for how essential oils produce a calming effect on the gut.

In summary, our study provides experimental evidence showing antispasmodic properties of VPEO. Future in vivo experiments evaluating the effect of VPEO on the intestinal microbiota or determining optimal dosage are required to obtain more clues into the potential use of VPEO as a new pharmacological approach for treating muscle spasms.

## 5. Conclusions

The findings obtained from this study on *Valeriana pilosa* essential oil (VPEO) unequivocally demonstrated its spasmolytic and antispasmodic effects in rat ileum. In silico and pharmacological analyses suggest that VPEO exerts its actions through blockade of muscarinic receptors and calcium channels. Collectively, these outcomes provide molecular insights that shed light on the traditional utilization of this plant for the treatment of gastrointestinal disorders.

## Figures and Tables

**Figure 1 pharmaceutics-15-02072-f001:**
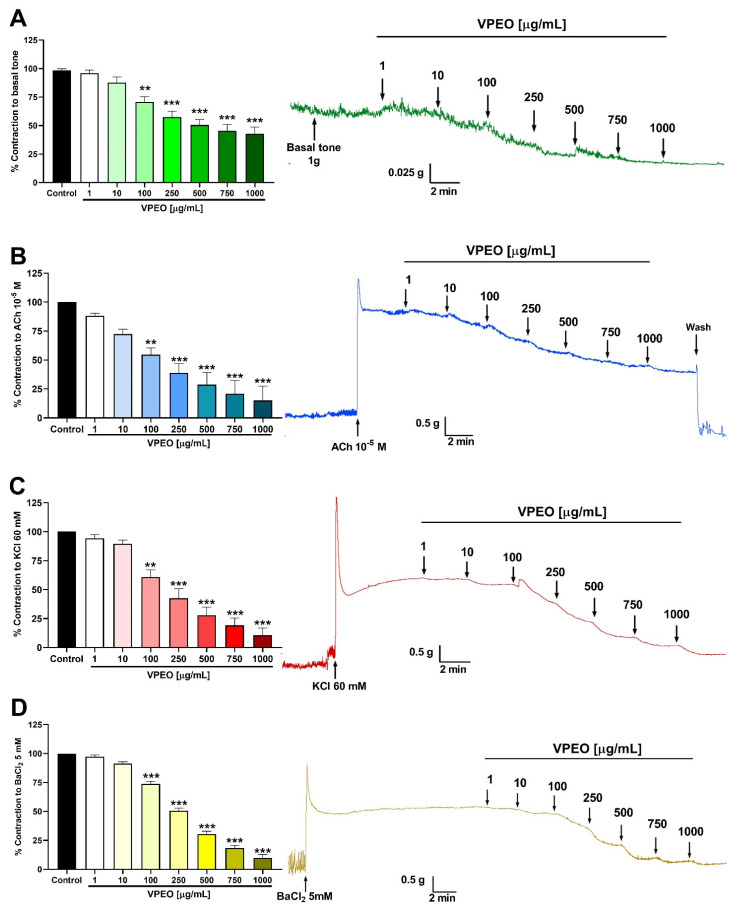
Effect of essential oil from *Valeriana pilosa* (VPEO) on rat ileum. VPEO relaxes the basal tone in rat ileum segments (**A**), VPEO-relaxed ileal segments precontracted with 10^−5^ M ACh (**B**), 60 mM KCl (**C**), or 5 mM BaCl_2_ (**D**). In panel (**A**), the control represents the basal tone (100% contraction) without any treatment. In panels (**B**–**D**), the control represents the maximum response (100%) induced by ACh, KCl, and BaCl_2_, respectively. In addition, original records of the relaxation effects of VPEO in the rat ileum are shown on the right side. Each bar represents the mean value of the response as a percentage ± SEM of five experiments (n = 5). Statistical differences: ** *p* < 0.01; *** *p* < 0.001 vs. control.

**Figure 2 pharmaceutics-15-02072-f002:**
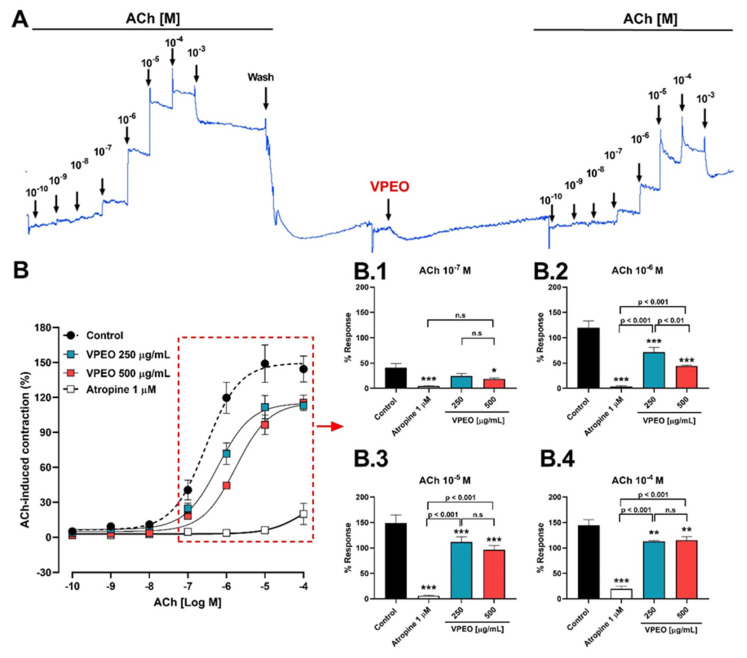
The antispasmodic activity of *Valeriana pilosa* essential oil (VPEO) reduces the contractile response to acetylcholine (ACh). The original record shows the effects of ACh on rat ileum segments in the absence and presence of VPEO. (**A**) The dose–response curve versus ACh when the rat ileum muscle tissue was pre-incubated with VPEO (250 and 500 μg/mL) and atropine 1 μM for 20 min before contraction with ACh. (**B**) Pre-incubation with VPEO (250 and 500 μg/mL) and atropine 1 μM reduced the contraction induced by Ach 10^−7^–10^−4^ M (**B.1**–**B.4**). Each point and bar represent the mean of maximal response in percentage ± SEM of five experiments (n = 5). * *p* < 0.05; ** *p* < 0.01; *** *p* < 0.001 vs. control.

**Figure 3 pharmaceutics-15-02072-f003:**
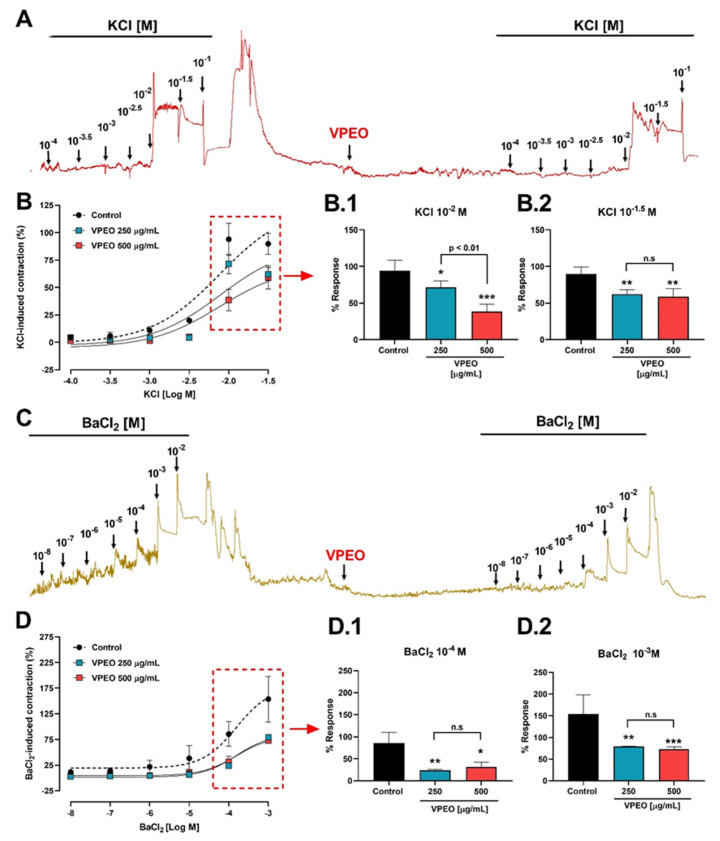
Antispasmodic activity of *Valeriana pilosa* essential oil (VPEO) in rat ileum. The original record shows the effects of KCl (**A**) and BaCl_2_ (**C**) on rat ileum in the absence and presence of VPEO. Rat ileum muscle tissue was pre-incubated with VPEO (250 and 500 μg/mL) for 20 min before contraction with KCl (**B**) and BaCl_2_ (**D**), and pre-incubation with VPEO 250 (**B.1**,**D.1**) and 500 μg/mL (**B.2**,**D.2**) reduced contraction. Each point represents the mean of maximal response in percentage ± SEM of five experiments (n = 5). n.s = not significant, statistical differences: * *p* < 0.05; ** *p* < 0.01; *** *p* < 0.001 vs. control.

**Figure 4 pharmaceutics-15-02072-f004:**
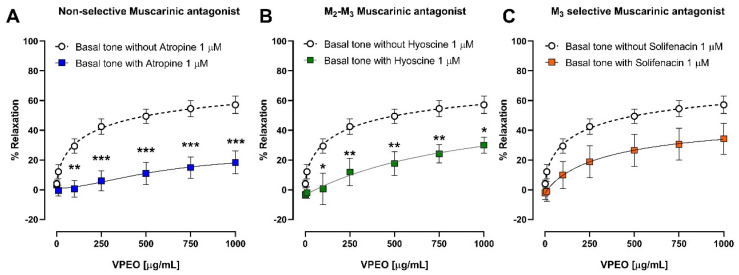
Relaxant effects of VPEO against (**A**) atropine 1 μM, (**B**) Hyoscine butylbromide 1 μM, (**C**) Solifenacin 1 μM in rat ileum sections. Each point represents the mean of maximal response in percentage ± SEM of five experiments (n = 5). n.s = not significant, statistical differences: * *p* < 0.05; ** *p* < 0.01; *** *p* < 0.001 vs. control.

**Figure 5 pharmaceutics-15-02072-f005:**
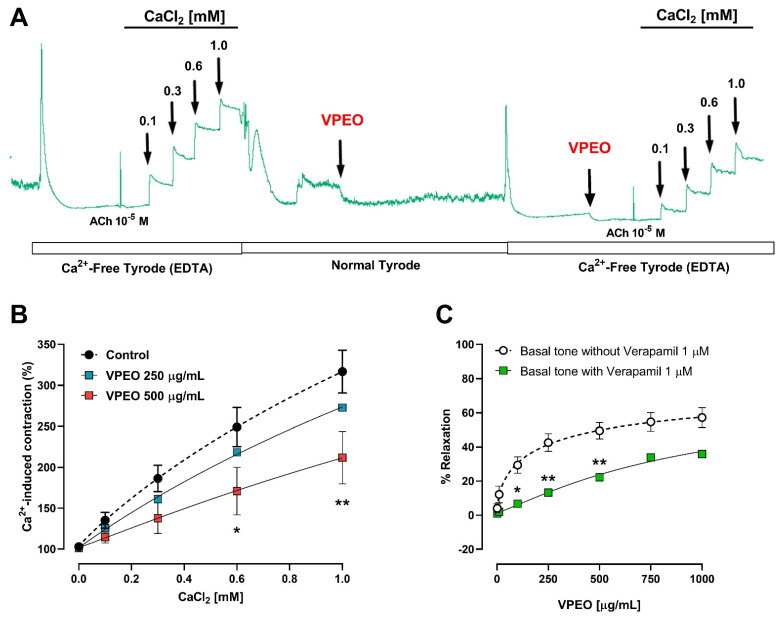
Effect of extracellular calcium in the presence and absence of VPEO in ileum rats precontracted with ACh (10^−5^ M). The original record shows the time course of the contractile response to CaCl_2_ in Ca^2+^-free Tyrode’s solution (containing 0.1 mM EDTA) in the absence and presence of VPEO (**A**). Concentration–response curve of CaCl_2_ in ileum rats in the absence and in the presence of VPEO (250 and 500 µg/mL) (**B**). Effects of verapamil (voltage-gated calcium channel blocker) on VPEO-induced relaxation in ileum rats (**C**). Each point represents the mean of maximal response in percentage ± SEM (n = 5–9). * *p* < 0.05; ** *p* < 0.01 vs. control.

**Figure 6 pharmaceutics-15-02072-f006:**
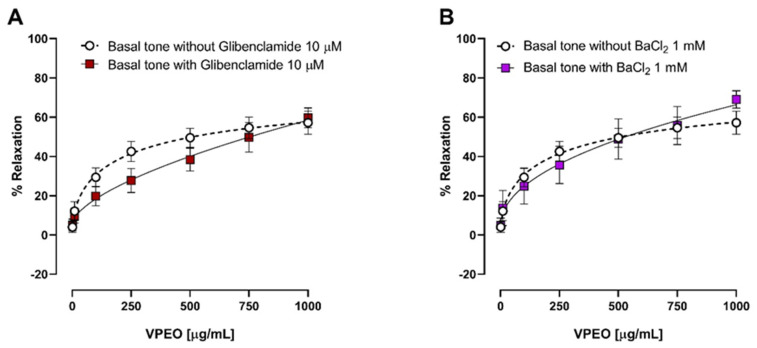
The effects of two potassium channel blockers separately of VPEO induced relaxation on rat ileum precontracted with KCl (20 mM) and ACh (10^−5^ M). (**A**) glibenclamide an ATP sensitive K⁺ channel blocker, 10 μM and (**B**) barium chloride, inward rectifier K⁺ channel blocker, 1 mM. Data represented as mean ± SEM (n = 4–8).

**Figure 7 pharmaceutics-15-02072-f007:**
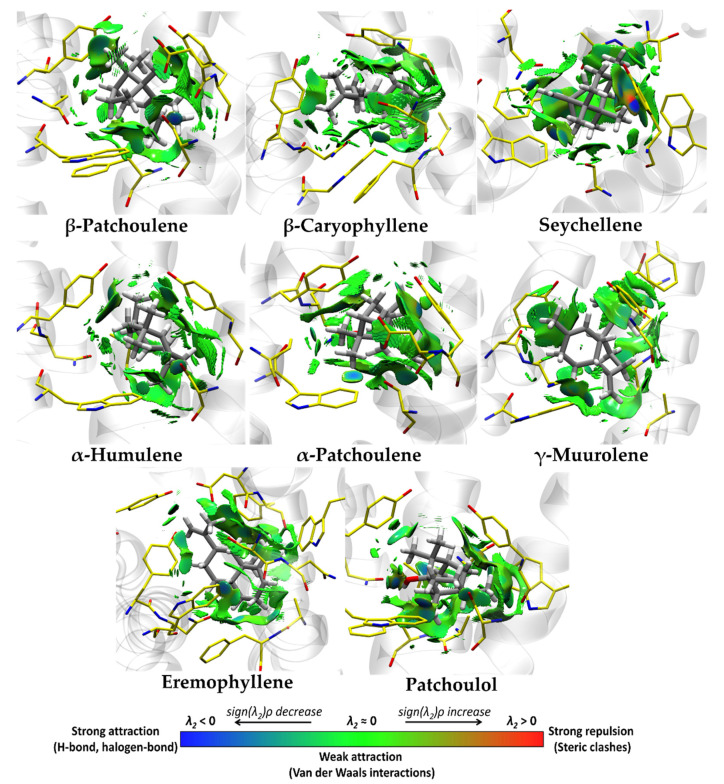
Non-covalent interactions analysis for the best four compounds (β-patchoulene, β-caryophyllene, γ-muurolene, and eremophyllene,), and major constituents of the molecular docking of VPEO (α-patchoulene, α-humulene, seychellene, patchoulol) bound to M_2_ muscarinic acetylcholine receptor.

**Figure 8 pharmaceutics-15-02072-f008:**
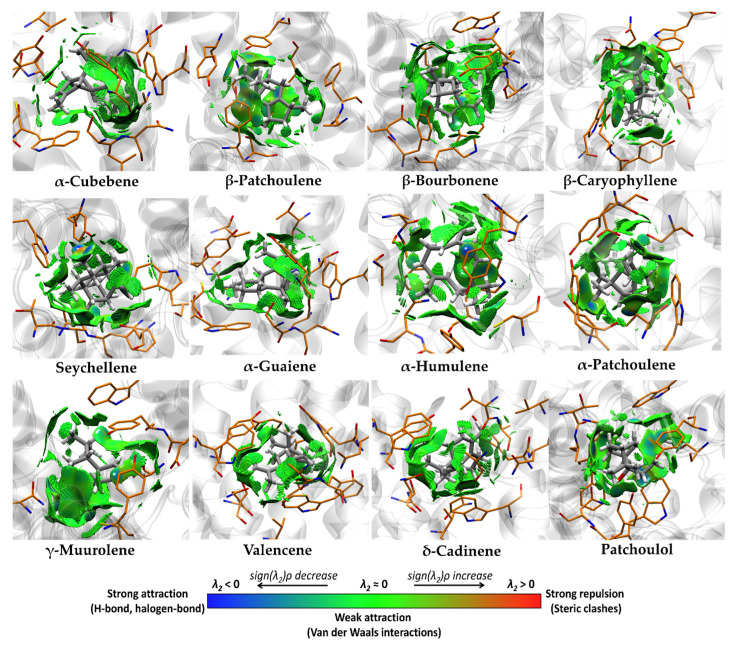
Non-covalent interactions analysis for the best nine compounds (α-cubebene, β-patchoulene, β-bourbonene, β-caryophyllene, α-guaiene, α-humulene, γ-muurolene, valencene, and δ-cadinene), and major (α-patchoulene, α-humulene, seychellene, patchoulol) constituents of the molecular docking of VPEO bound to M_3_ muscarinic acetylcholine receptor.

**Table 1 pharmaceutics-15-02072-t001:** Cartesian coordinates of the center grid box (in Å) for M_2_ Muscarinic Acetylcholine Receptor, M_3_ Muscarinic Acetylcholine Receptor, and Ca_v_1.2 (L-type voltage-gated calcium channel).

Protein	PDBID	Center Grid Box		
		x	y	z
M_2_ Muscarinic Acetylcholine Receptor	3UON	7.79	0.25	−3.95
M_3_ Muscarinic Acetylcholine Receptor	4DAJ	−14.79	−7.50	−42.53
Ca_v_1.2 L-type voltage-gated calcium channel	5V2P	7.76	42.10	123.95

**Table 2 pharmaceutics-15-02072-t002:** Heat map of the intermolecular docking energy values (kcal·mol^−1^) of VPEO compounds on M_2_ and M_3_ Muscarinic Acetylcholine and Ca_v_1.2 (L-type VGCC) receptors.

N°	Components	M_2_	M_3_	Ca_v_1.2
1	Isovaleric acid	−4.8	−4.5	−4.2
2	Tricyclene	−6.8	−6.5	−4.4
3	α-Thujene	−6.5	−6.7	−4.6
4	α-Pinene	−6.5	−6.8	−4.3
5	Camphene	−6.9	−6.7	−4.6
6	3-Methyl valeric acid	−4.7	−4.8	−4.4
7	Sabinene	−6.5	−6.8	−4.7
8	1-Octen-3-ol	−5.2	−5.1	−3.7
9	β-Pinene	−6.6	−6.8	−4.4
10	Myrcene	−6	−5.9	−3.7
11	Limonene	−6.5	−6.5	−4.5
12	p-Cymene	−6.6	−6.5	−4.6
13	1,8-Cineole	−6.7	−6.7	−4.4
14	Linalool	−6	−5.9	−4.5
15	Isopentyl isovalerate	−5.8	−6.0	−4.3
16	Camphor	−6.9	−6.7	−4.2
17	Menthone	−6.9	−6.7	−4.4
18	Isomenthone	−6.9	−6.7	−4.4
19	Borneol	−6.8	−6.5	−4.1
20	Neomenthol	−6.8	−6.6	−4.2
21	Menthol	−6.4	−6.3	−4.2
22	Carvone	−6.9	−6.9	−4.7
23	Menthyl acetate	−7.5	−7.4	−4.7
24	α-Cubebene	−8.5	−9.0	−5.5
25	Cyclosativene	−8.5	−8.9	−5.4
26	α-Copaene	−8.2	−8.2	−5.3
27	β-Patchoulene	−9.1	−9.4	−5.8
28	β-Bourbonene	−8.9	−9.1	−5.1
29	β-Elemene	−8.1	−8.0	−5
30	β-Caryophyllene	−9.0	−9.1	−5.6
31	Seychellene	−7.9	−8.4	−5.4
32	α-Guaiene	−8.6	−9.0	−5.7
33	α-Humulene	−8.6	−9.1	−5.3
34	allo-Aromadendrene	−8.3	−8.6	−5.4
35	α-Patchoulene	−8.3	−8.7	−5.5
36	γ-Muurolene	−9.1	−9.2	−5.5
37	Germacrene-D	−8.6	−8.8	−5.2
38	Valencene	−8.5	−9.0	−5.6
39	Eremophyllene	−9.1	−8.9	−5.5
40	γ-Cadinene	−8.6	−8.6	−5.5
41	7-epi-α-Selinene	−8.3	−8.8	−5.4
42	δ-Cadinene	−8.8	−9.0	−5.4
43	Spathulenol	−8.6	−8.6	−5.4
44	β-Caryophyllene oxide	−8.1	−8.3	−5.5
45	T-Cadinol	−8.5	−8.8	−5.4
46	δ-Cadinol	−8.4	−8.8	−5.4
47	Patchoulol	−7.8	−8.6	−5.2

Values are listed as a three-colored scheme from red (high energy) to green (low energy).

**Table 3 pharmaceutics-15-02072-t003:** Average molecular docking results for the 47 compounds of VPEO regarding the M_2_ and M_3_ Muscarinic Acetylcholine Receptor and Ca_v_1.2 (L-type VGCC). Intermolecular docking energy values (∆*E_binding_*), Ligand Efficiency (*LE*), and normalizing binding energy (*IE_norm.binding_*).

Proteins	$\bar{X}$∆*E_binding_* (kcal·mol^−1^)	$\bar{X}$*LE* (kcal·mol^−1^)	$\bar{X}$*IE_norm.binding_* (kcal·mol^−1^)
M_2_ Muscarinic Acetylcholine Receptor	−7.67	0.55	−2.06
M_3_ M_3_ Muscarinic Acetylcholine Receptor	−8.03	0.58	−2.15
Ca_v_1.2 (L-type VGCC)	−5.11	0.37	−1.37

**Table 4 pharmaceutics-15-02072-t004:** Molecular docking results from the best compounds of VPEO regarding the M_2_ Muscarinic Acetylcholine Receptor and M_3_ Muscarinic Acetylcholine Receptor with the major constituents of VPEO. Intermolecular docking energy values (∆*E_binding_*), K_d_ values, Ligand Efficiency (*LE*), Binding Efficiency Index (*BEI*), Lipophilic Ligand Efficiency (*LLE*), and normalizing binding energy (*IE_norm,binding_*) for the complexes.

N°	Components	∆*E_binding_* (kcal·mol^−1^)		K_d_		*LE* (kcal·mol^−1^)		*BEI* (kDa)		*LLE*		*IE_norm.binding_* (kcal·mol^−1^)	
		M_2_	M_3_	M_2_	M_3_	M_2_	M_3_	M_2_	M_3_	M_2_	M_3_	M_2_	M_3_
24	α-Cubebene	−8.5	−9.0	5.90 × 10^−7^	2.54 × 10^−7^	0.60	0.60	30.5	32.3	2.0	2.3	−2.2	−2.3
27	β-Patchoulene	−9.1	−9.4	2.14 × 10^−7^	1.29 × 10^−7^	0.61	0.63	32.6	33.7	2.1	2.3	−2.3	−2.4
28	β-Bourbonene	−8.9	−9.1	3.00 × 10^−7^	2.14 × 10^−7^	0.59	0.61	31.9	32.6	2.3	2.4	−2.3	−2.3
30	β-Caryophyllene	−9.0	−9.1	2.54 × 10^−7^	2.14 × 10^−7^	0.60	0.61	32.0	32.3	1.8	1.9	−2.3	−2.3
31	Seychellene *	−7.9	−8.4	1.62 × 10^−6^	6.98 × 10^−7^	0.53	0.56	28.3	30.1	1.4	1.7	−2.0	−2.2
32	α-Guaiene	−8.6	−9.0	4.98 × 10^−7^	2.54 × 10^−7^	0.60	0.60	30.8	32.3	1.6	1.9	−2.2	−2.3
33	α-Humulene *	−8.6	−9.1	4.98 × 10^−7^	2.14 × 10^−7^	0.57	0.61	30.8	32.6	1.3	1.6	−2.2	−2.3
35	α-Patchoulene *	−8.3	−8.7	8.26 × 10^−7^	4.21 × 10^−7^	0.55	0.58	29.8	31.2	1.7	2.0	−2.1	−2.2
36	γ-Muurolene	−9.1	−9.2	2.14 × 10^−7^	1.81 × 10^−7^	0.61	0.61	32.6	33.0	2.1	2.2	−2.3	−2.4
38	Valencene	−8.5	−9.0	5.90 × 10^−7^	2.54 × 10^−7^	0.60	0.60	30.5	32.3	1.5	1.9	−2.2	−2.3
39	Eremophyllene	−9.1	−8.9	2.14 × 10^−7^	3.00 × 10^−7^	0.60	0.60	32.6	31.9	1.9	1.8	−2.3	−2.3
42	δ-Cadinene	−8.8	−9.0	3.55 × 10^−7^	2.54 × 10^−7^	0.60	0.60	31.6	32.3	1.7	1.9	−2.3	−2.3
47	Patchoulol *	−7.8	−8.6	1.92 × 10^−6^	4.98 × 10^−7^	0.49	0.54	25.7	28.3	2.1	2.7	−2.0	−2.2

* Represent the major constituents of the VPEO.

**Table 5 pharmaceutics-15-02072-t005:** Amino acid residues of M_2_ Muscarinic Acetylcholine Receptor (M_2_R) and hydrogen bonding with the VPEO molecules within a distance of 3.5 Å.

	Interacting Amino Acids in the Binding Pocket of M_2_R
Compound	Amino Acids (Distance in Å)
β-Patchoulene	Ala194 (5.43), Cys429 (4.53), Tyr104 (3.57), Phe195 (5.17), Trp400 (4.62), Tyr403 (3.77), Tyr426 (4.49).
β-Caryophyllene	Ala194 (3.83), Tyr104 (4.57), Trp155 (4.66), Trp400 (5.29), Tyr403 (4.51).
Seychellene *	Ala191 (4.22), Ala194 (4.77), Tyr104 (3.99), Trp155 (4.75), Phe181 (4.97), Trp400 (4.90), Tyr403 (4.86).
α-Humulene *	Ala194 (5.44), Tyr104 (4.80), Trp400 (5.20), Tyr403 (4.78).
α-Patchoulene *	Cys429 (5.17), Tyr104 (5.14), Phe181 (5.46), Trp400 (4.90), Tyr403 (5.38).
γ-Muurolene	Ala191 (5.19), Ala194 (4.98), Cys429 (5.20), Tyr104 (5.04), Phe181 (4.80), Phe195 (5.25), Trp400 (5.20), Tyr403 (4.43), Tyr426 (4.70).
Eremophyllene	Val111 (5.15), Ala194 (3.92), Tyr104 (5.15), Trp155 (4.26), Trp400 (5.09), Tyr403 (4.12), Tyr426 (4.44).
Patchoulol *	Asn404 (2.08) **, Ala191 (3.97), Ala194 (4.31), Tyr104 (4.26), Trp155 (4.18), Phe195 (4.82), Trp400 (4.86), Tyr403 (4.90).

* Represent the major constituents of the VPEO. ** Represent H-bonds.

**Table 6 pharmaceutics-15-02072-t006:** Amino acid residues of M_3_ Muscarinic Acetylcholine Receptor (M_3_R) and hydrogen bonding with the VPEO molecules within a distance of 3.5 Å.

	Interacting Amino Acids in the Binding Pocket of M_3_R
Compound	Amino Acids (Distance in Å)
α-Cubebene	Trp503 (3.65), Ala235 (4.43), Ala238 (4.30), Cys532 (3.89), Tyr148 (5.34), Trp199 (4.60), Trp503 (4.59), Tyr506 (4.85).
β-Patchoulene	Tyr506 (2.76), Ala238 (3.96), Cys532 (4.04), Tyr148 (3.64), Phe239 (5.47), Trp503 (4.39), Tyr506 (5.02).
β-Bourbonene	Ala235 (4.36), Ala238 (4.16), Cys532 (4.77), Tyr148 (5.22), Trp503 (5.24), Tyr506 (4.78).
β-Caryophyllene	Ala238 (4.70), Val155 (4.08), Tyr148 (5.18), Trp199 (4.23), Trp503 (4.88), Tyr506 (4.25), Tyr529 (5.34).
Seychellene *	Ala235 (4.08), Ala238 (3.79), Tyr148 (4.35), Trp199 (4.21), Trp503 (4.04), Tyr506 (4.77).
α-Guaiene	Trp503 (3.71), Ala235 (3.68), Ala238 (5.23), Cys532 (4.46), Val510 (4.30), Tyr148 (5.06), Trp199 (5.05), Trp503 (4.47), Tyr506 (4.23).
α-Humulene *	Ala235 (3.97), Cys532 (3.57), Val155 (4.38), Trp503 (3.59), Tyr506 (3.87), Tyr529 (4.68).
α-Patchoulene *	Tyr506 (2.57), Cys532 (4.03, Tyr148 (4.16), Trp503 (4.19), Tyr529 (4.19).
γ-Muurolene	Cys532 (4.01), Tyr148 (3.73), Trp199 (5.01), Trp503 (4.31), Tyr506 (4.37), Tyr529 (4.69).
Valencene	Ala238 (3.94), Cys532 (4.43), Val155 (4.14), Tyr148 (4.10), Trp199 (4.40), Trp503 (5.17), Tyr506 (4.32), Tyr529 (4.76).
δ-Cadinene	Ala235 (3.67), Ala238 (4.65), Cys532 (3.62), Tyr148 (5.36), Trp199 (5.46), Trp503 (4.34), Tyr506 (5.01).
Patchoulol *	Asn507 (2.58) **, Ala235 (3.83), Ala238 (3.61), Tyr148 (4.54), Trp199 (4.64), Phe239 (5.23), Trp503 (4.40), Tyr506 (4.66).

* Represent the major constituents of the VPEO. ** Represent H-bonds.

## Data Availability

Not applicable.

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
