# Peer review of "Antispasmodic Effect of *Valeriana pilosa* Root Essential Oil and Potential Mechanisms of Action: Ex Vivo and In Silico Studies"

_pharmaceutics, 2023, doi:10.3390/pharmaceutics15082072_

Round 1
Reviewer 1 Report
Dear author,
After the review process, I have several comments: all Materials and Methods sections should include references; in the introduction, the first paragraph should be expanded with new findings about microbiota bioactivity and bioavailability of functional compounds; starting from the phytochemical analysis of essential oils, the discussion should include comments related to the bioavailability process, that could influence the activity in the case of extract.
Best regards!
Author Response
We are grateful to the referee for the suggestions. In the revised version, we have incorporated their recommendations.

Reviewer 2 Report
The study investigates the potential of Valeriana pilosa root essential oil as a treatment for muscle spasms. The research is conducted ex vivo and in silico, and the results suggest that the essential oil has antispasmodic effects and may act through multiple mechanisms.
There are several issues that need to be noted:
1. the given texts do not provide enough information to evaluate the quality of the research methodology and data analysis. such as sample randomization, blinding. Without this information, it is difficult to evaluate the quality of the study and the reliability of the results.
2. the method is that the study was conducted ex vivo and in silico, which means that the results may not accurately reflect the effects of Valeriana pilosa root essential oil in living organisms. In vivo studies would be necessary to confirm the antispasmodic effects of the essential oil and its possible mechanisms of action.
3. the study does not provide a clear discussion of the practical implications of the research, such as the potential for developing new treatments for muscle spasms. Without this discussion, it is unclear how the research could be translated into clinical practice and benefit patients with muscle spasms.
It is important to note that further research is needed to confirm these results and to determine the optimal dosage and administration of the essential oil. Overall, this paper presents a valuable contribution to the field of natural remedies for muscle spasms and warrants further investigation.
The manuscript needs to be appropriately polished in English.
Author Response
Reviewer
The study investigates the potential of Valeriana pilosa root essential oil as a treatment for muscle spasms. The research is conducted ex vivo and in silico, and the results suggest that the essential oil has antispasmodic effects and may act through multiple mechanisms.
There are several issues that need to be noted:
- the given texts do not provide enough information to evaluate the quality of the research methodology and data analysis. such as sample randomization, blinding. Without this information, it is difficult to evaluate the quality of the study and the reliability of the results.
Answer: We thank the reviewer for the comments. The experiments were neither randomized nor blinded. Both control and treated experiments were performed in the same days.
- the method is that the study was conducted ex vivo and in silico, which means that the results may not accurately reflect the effects of Valeriana pilosa root essential oil in living organisms. In vivo studies would be necessary to confirm the antispasmodic effects of the essential oil and its possible mechanisms of action.
Answer: We agree that in vivo experiments are critical to evaluate the potential use of VPEO as treatment. We have included In the Discussion section a paragraph highlighting the relevance of in vivo experiments in future studies.
- the study does not provide a clear discussion of the practical implications of the research, such as the potential for developing new treatments for muscle spasms. Without this discussion, it is unclear how the research could be translated into clinical practice and benefit patients with muscle spasms.
It is important to note that further research is needed to confirm these results and to determine the optimal dosage and administration of the essential oil. Overall, this paper presents a valuable contribution to the field of natural remedies for muscle spasms and warrants further investigation.
Answer: We have included In the Discussion section a paragraph including the suggestion.
